# Context-encoding Variational Autoencoder
# for Unsupervised Anomaly Detection

**David Zimmerer**[1]
**Simon Kohl**[1]
**Jens Petersen**[1]
**Fabian Isensee**[1]
**Klaus Maier-Hein**[1]
[1] *German Cancer Research Center (DKFZ), Heidelberg, Germany*

## Abstract

Unsupervised learning can leverage large-scale data sources without the need for annotations. In this context, deep learning-based autoencoders have shown great potential in detecting anomalies in medical images. However, especially Variational Autoencoders (VAEs) often fail to capture the high-level structure in the data. We address these shortcomings by proposing the context-encoding Variational Autoencoder (ceVAE), which improves both, the sample- as well as pixelwise results. In our experiments on the BraTS-2017 and ISLES-2015 segmentation benchmarks the ceVAE achieves unsupervised AUROCs of 0.95 and 0.89, respectively, thus outperforming other reported deep-learning based approaches.

## 1. Introduction

In the last years several computer-aided diagnosis systems have reported near human-level performance (Liu et al., 2017; Gulshan et al., 2016). However those approaches are only applicable in a narrow range of cases, where larger amounts of annotated training data is available. Unsupervised approaches on the other hand do not need any annotated data and thus allow broader applicability and independence of human errors in the annotation. Recently, several deep-learning based pixelwise anomaly detection approaches have shown great promise (Abati et al., 2018; Baur et al., 2018; Chen et al., 2018; Chen and Konukoglu, 2018; Pawlowski et al., 2018; Schlegl et al., 2017). Most of these approaches are built on the reconstruction error of generative models, Variational Autoencoders (VAEs) in particular. However recent research has critiqued VAEs for their low capability to extract high-level structure in the data, and suggested improvements over the base model (Nalisnick et al., 2018; Zhao et al., 2017; Maaløe et al., 2019). Here we use context-encoding (Pathak et al., 2016) to steer the VAE towards learning more discriminative features. Our results suggest that these features can improve out-of-distribution/anomaly-detection tasks and as such aid VAEs in capturing the data distribution.

## 2. Methods

VAE-based methods are, next to flow-based and autoregressive models, one of the most commonly used models for density estimation and out-of-distribution/anomaly-detection

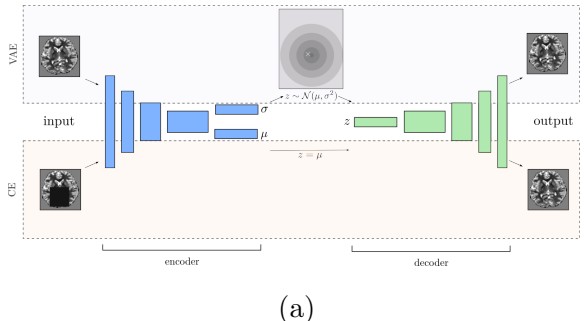 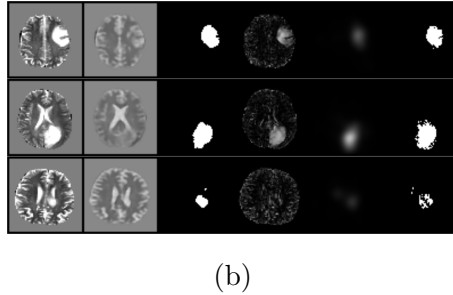

(a)            (b)

Figure 1: (a) ceVAE model structure. (b) 3 Samples from the BraTS dataset. Shown from left to right is the input, the reconstruction, the ground-truth annotation, the reconstruction-error, the KL-Gradient, the final resulting segmentation

tasks. Denoising autoencoders and especially context-encoders (CE) on the other hand were shown to learn discriminative features, invariant to perturbations of the data (Pathak et al., 2016; Vincent et al., 2010). We propose to combine them into a context-encoding VAE (ceVAE), as shown in Fig. 1a, to learn more discriminative features and to prevent posterior collapse in VAEs. Each data sample $x$ is used to train a normal VAE. At the same time, CE-noise perturbed data samples $\tilde{x}$ are used to train a CE, where we do not sample from the latent posterior distribution but use the mean as encoding $z$. This results in the following objective:

$$\min L_{ceVAE} = (1 - \lambda)[L_{KL}(f_\mu(x), f_\sigma(x)^2) + L_{rec_{VAE}}(x, g(z))] + \lambda L_{rec_{CE}}(x, g(f_\mu(\tilde{x}))), \quad (1)$$

with neural network encoders $f_\mu$, $f_\sigma$ and decoder $g$. $\lambda$ weighs the VAE objective against the CE objective (default $\lambda = 0.5$).

To detect samplewise anomalies we use the approximated evidence lower bound (ELBO) from the VAE as proxy for the data likelihood and consequently as anomaly score. Since a low ELBO can be either caused by a high reconstruction error $L_{rec}$ or a high KL-divergence $L_{KL}$, we use both for a pixelwise anomaly score. As commonly done (Baur et al., 2018; Chen et al., 2018; Pawlowski et al., 2018), we use the reconstruction error directly as pixelwise indication for anomalies. To complement this with a pixelwise anomaly score from the KL-divergence, we use the derivative of the KL-divergence with respect to the input $\frac{\partial L_{KL}}{\partial x}$, which was previously reported to show good results on its own (Zimmerer et al., 2018). We combine the two pixelwise scores using pixelwise multiplication:

$$\mathcal{A}_{pixel} = |x - g(f(x))| \odot |\frac{\partial L_{KL}(x)}{\partial x}|. \quad (2)$$

## 3. Experiments & Results

We trained the model on 2D slices of T2-weighted images from the HCP dataset (Van Essen et al., 2012) (N=1092) to learn the distribution of healthy patient images. The model was tested to find anomalies in the BraTS2017 (Bakas et al., 2017; Menze et al., 2015) (N=266) and the ISLES2015 (Maier et al., 2017) (N=20) dataset. For a samplewise detection slices

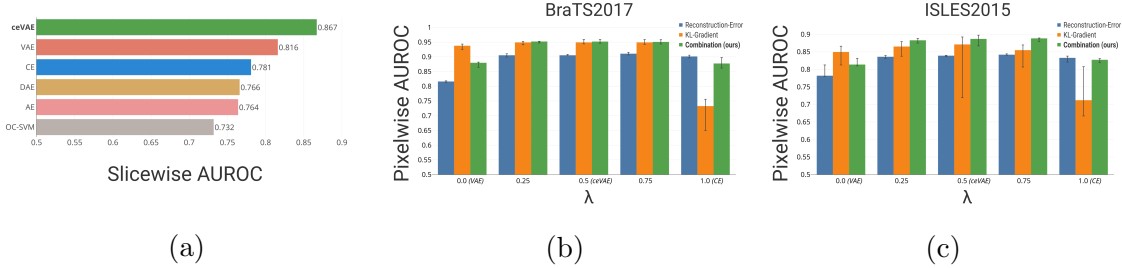

Figure 2: (a) Slicewise results for the BraTS dataset. (b,c) Pixelwise results for the BraTS2017 (b) and ISLES2015 (c) dataset.

without any annotations were regarded as normal and samples with $> 20$ annotated pixels were regarded as anomalies. For the pixelwise detection, annotated pixels were regarded as anomalies, while pixels without any annotations were regarded as normal. For the encoder and decoder networks, we chose fully convolutional networks with five 2D-Conv-Layers and 2D-Transposed-Conv-Layers respectively with kernel size 4 and stride 2, each layer followed by a LeakyReLU non-linearity. The models were trained with Adam with a learning rate of $2 \times 10^{-4}$ and a batch size of 64 for 60 epochs.

The samplewise results can be seen in Fig. 2a, indicating that the CE-objective can aid VAEs for anomaly detection. To analyze the posterior collapse we inspect the 0.95 quantile of the KL-divergence over the test samples after training. For the ceVAE $L_{KL}$ the 0.95 quantile is 2.93 while for the VAE it is 0.53, indicating that more latent variables are used in the ceVAE and fewer have collapsed.

The pixelwise results can be seen in Fig. 2b,c. The combined approach in Eq. 2 as well as its two components are presented individually. Including the CE objective into the VAEs appears to already improve the pixelwise reconstruction-only based detection. The same can also be observed for the KL-Gradient based detection. The best result on the BraTS and ISLES dataset are achieved using $\lambda = 0.5$ for the model proposed in Eq. 1. Using 20% of the BraTS2017 testset to determine a threshold, and calculating the Dice score of the other 80% of the testset, results in a Dice score of 0.52 outperforming previously reported deep-learning based anomaly detection results on similar datasets (Chen et al., 2018). While the analysis was performed on $64 \times 64$-sized images, early results with images sizes of $192 \times 192$ show a better performance, in particular on the ISLES2015 dataset with an AUROC of 0.92.

## 4. Discussion & Conclusion

We presented a way to integrate context encoding into VAEs. This leads to a better utilization of the latent-variables and shows better performance detecting out-of-distribution samples/anomalies samplewise and pixelwise. We are confident that this represents an important step towards anomaly detection without the need for annotated data and thus make computer-aided diagnosis applicable to a wider variety of cases.

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
