# OpenReview forum: "Context-encoding Variational Autoencoder for Unsupervised Anomaly Detection"
_MIDL.io/2019/Conference/Abstract — MIDL Abstract 2019_

### Official Review · AnonReviewer1 · 2019-04-29
**Big improvement by combining two existing methods**

**Rating:** 3
**Confidence:** 2

**Review:**

By combining both VAE and CE, the anomaly detection performance is dramatically improved, to a surprising extent. Despite the interesting result, there is little explanation for that. It would be interesting to analyze the difference between “z” from CE-augmented one and the healthy one. In addition, the improvement from CE and VAE individually should be compared visually.

---

### Official Review · AnonReviewer2 · 2019-04-30
**Interesting approach for an important and difficult problem**

**Rating:** 3
**Confidence:** 2

**Review:**

This manuscript addresses the important and difficult problem of unsupervised segmentation of pathology via anomaly detection. It explores the novel idea of combining context-encoders and variational autoencoders and demonstrates that this results in a benefit, which is explored further by varying the relative weights of the two components in the loss.

Results seem promising, even though I would have liked to see more information concerning the importance of image registration. Were the data used in this study pre-normalized to some standard space? Did the authors co-register them? Would the network be able to deal with completely arbitrary head orientations?

Figures are very small and only became legible after zooming in greatly on the electronic version.

---

### Decision · Program_Chairs · 2019-05-06
**Acceptance Decision**

Accept